# Fermentation of Jamaican Cherries Juice Using *Lactobacillus plantarum* Elevates Antioxidant Potential and Inhibitory Activity against Type II Diabetes-Related Enzymes

**DOI:** 10.3390/molecules26102868

**Published:** 2021-05-12

**Authors:** Andri Frediansyah, Fitrio Romadhoni, Rifa Nurhayati, Anjar Tri Wibowo

**Affiliations:** 1Research Division for Natural Products Technology (BPTBA), Indonesian Institute of Sciences (LIPI), Yogyakarta 55861, Indonesia; rifa004@gmail.com; 2Department of Chemistry, Faculty of Mathematic and Natural Science, Islamic University of Indonesia (UII), Yogyakarta 55584, Indonesia; kimiajogja2@gmail.com; 3Department of Biology, Faculty of Science and Technology, Sunan Kalijaga Islamic State University (UIN Sunan Kalijaga), Yogyakarta 55281, Indonesia; lactobacilluspentosus@gmail.com; 4Departement of Biology, Faculty of Science and Technology, Airlangga University, Kampus C, Mulyorejo, Surabaya 60115, Indonesia; 5Biotechnology of Tropical Medicinal Plants Research Group, Airlangga University, Kampus C, Mulyorejo, Surabaya 60115, Indonesia

**Keywords:** Jamaican cherry, fermentation, *Lactobacillus plantarum*, antioxidant, antidiabetic, food nutrition improvement

## Abstract

Jamaican cherry (*Muntinga calabura* Linn.) is tropical tree that is known to produce edible fruit with high nutritional and antioxidant properties. However, its use as functional food is still limited. Previous studies suggest that fermentation with probiotic bacteria could enhance the functional properties of non-dairy products, such as juices. In this study, we analyze the metabolite composition and activity of Jamaican cherry juice following fermentation with *Lactobacillus plantarum* FNCC 0027 in various substrate compositions. The metabolite profile after fermentation was analyzed using UPLC-HRMS-MS and several bioactive compounds were detected in the substrate following fermentation, including gallic acid, dihydrokaempferol, and 5,7-dihydroxyflavone. We also found that total phenolic content, antioxidant activities, and inhibition of diabetic-related enzymes were enhanced after fermentation using *L. plantarum*. The significance of its elevation depends on the substrate composition. Overall, our findings suggest that fermentation with *L. plantarum* FNCC 0027 can improve the functional activities of Jamaican cherry juice.

## 1. Introduction

Jamaican cherry (*Muntingia calabura* Linn.) is a plant species belonging to Elaeocarpaceae. This plant is indigenous to Southern Mexico, Northern South America, Central America, Trinidad, St. Vincent, and the Greater Antilles [1]. It is also spread and commonly found in other tropical countries including Indonesia, Malaysia, the Philippines, and India [2]. The flower, roots, and barks are used as folk remedies for various medical conditions including fever, liver disease, incipient cold, and antiseptic agents in Southeast Asia [1]. The leaves are known to contain phenolic compounds that exhibits various biological activities such as antimicrobial [3], antioxidant [4], anti-cancer [5], hepatoprotective [6], and hypotensive activities [7]. In addition, Jamaican cherry fruit is edible; they can be eaten raw or processed as juice. Morphologically, the fruit is relatively small, weighted around 1.50 g, round smooth-shape, and the color turns from green to red when ripened. It has a sweet taste due to low titratable acidity and high content of soluble solids. The fruit is also known to have high nutritional value [8] and contain high concentration of flavonoid and phenolic compounds [9]. Due to its taste, nutritional value, and high content of antioxidant compounds, Jamaican cherry fruit has a great potential to be used as a functional food in the food industry.

In previous work, we reported the isolation of lactic acid bacteria from Jamaican cherry fruit [10]. Despite their abundance and diversity, little is known about lactic acid bacteria natural physiological function. Nevertheless, these bacteria are widely known in the food industry due to their probiotic properties [11]. Fermentation using lactic acid bacteria was reported to enhance the flavor [12], shelf life [13], bioavailability [14], and functional quality of the fermented substrates [15]. Our previous study showed that fermentation of black grape juice using *Lactobacilli* could increase juices’ functional properties in vitro [16]. The use of *L. pentosus* and *Leuconostoc* in the fermentation of carrot juice also shown to improve the bioavailability of iron in the juice [17]. Another study showed that lactic bacteria and other associated microorganisms in kefir could elevate the α-glucosidase inhibitory effect of *Aronia melanocarpa* fruit juice [18]. The market interest and demands for functional food are continuously increasing [19], aforementioned studies suggest that probiotics bacteria could be used to enhance the functional properties of non-dairy products such as juices. In this study, we use *L. plantarum* strain FNCC 0027 for the fermentation of Jamaican cherry juice. Six different substrate compositions were tested and functional properties of the fermented juices in each of the substrate compositions were reported, including phenolic content, antioxidant capacity, and the inhibition of several type II diabetes-related enzymes. Further, we also provided the metabolite profile of the fermented juice. Here we showed that Jamaican cherry juice fermentation using *L. plantarum* FNCC 0027 can effectively improve functional properties of the juice.

## 2. Results and Discussion

### 2.1. Fermentation of Jamaican Juice by L. plantarum

Fermentation using various substrate compositions containing different ratios of Jamaican cherry juice and MRS broth (M1 to M6) was monitored for 48 h. As shown in Figure 1A. the initial viable count of *L. plantarum* was 7.02 log CFU/mL and the initial pH was 6.51. The rapid growth of *L. plantarum* was observed after 4 h of fermentation while the pH of the substrate continuously declined in an opposite trend. The viable count of *L. plantarum* reached a maximum at 24 h and was maintained until 48 h. The highest viable count observed was 9.3 logs CFU/mL at 24 h in M6, followed by M5, M4, M2, M1, and M3 with 8.9, 8.6, 8.4, 8.5, and 8.5 log CFU/mL, respectively (Figure 1A). Our data showed that the growth rate of *L. plantarum* was positively correlated with the amount of MRSB in the substrate. Propagation of viable cells and growth pattern are crucial for the production of desired metabolite in the fermented substrate. Østlie, et al. [20] reported that the production of organic acid, carbon dioxide, and volatile compounds was influenced by the total cell count of probiotic *L. acidophilus, L. rhamnosus, L. reuteri*, and *Bifidobacterium animalis* in the fermented substrate. On the other hand, *L. plantarum* viable count was negatively correlated with substrate pH, while the viable count is increasing the pH is decreasing over the fermentation period (*p* < 0.05) (Figure 1B). The pH value of the substrate is an essential parameter to assess the progress and endpoint of the fermentation; it was reported to influence the flavor of the final product [21]. After 48 h of fermentation, the pH of the fermented juices has decreased from 6.5 to 3.3 (M1), 3.4 (M2), 4 (M3), 3.7 (M4), 3.4 (M5), and 3.9 (M6). This pattern is quite similar to the 48-h-fermentation of Indonesian black grape juice using *L. plantarum* [16].

### 2.2. Total Phenolic Content after Fermentation

The fruit of Jamaican cherry are reported to be rich in phenol, flavonoid, and other bioactive substances [8]. Phenols are essential antioxidants in fruit and vegetables, therefore, in this study we evaluated total phenolic content in different substrates formulation (M1 to M6) before and after fermentation. Our data revealed that fermentation using *L. plantarum* can significantly increase total phenolic compounds in M1 and M2 formulation to about 1.3- and 1.4-folds (*p* < 0.05), respectively (Figure 2). This result is in consonance with published studies about fruit fermentation using *L. plantarum*, including in the fermentation of blueberry [22], mulberry [23], and kiwi juice [24]. Changes in total phenolic content in the substrate might occur through multiple factors. For example, changes in substrate pH during fermentation period could affect the structure of the phenolic compound [22]. Alternatively, a macromolecular form of phenol can be disintegrated into small phenols due to specific metabolic activity of the lactic acid bacteria, such as de-glycosylation [23]. *Lactobacillus* fermentation does not always have a positive effect on phenol concentration. Liao, et al. [25] reported that fermentation using *L. brevis* MPL39 can significantly reduce phenolic content in mango juice. Therefore, it should be considered that different lactic acid bacteria strains and substrate matrices could have different effects on the phenolic content.

### 2.3. Antioxidative Activity of the Fermented Juices

Antioxidants that present in fermented substrates, such as polyphenol and flavonoid, can eliminate free radicals through electron donation or hydrogen atom transfer. Various types of antioxidative compounds may act against oxidizing agents through diverse mechanisms [26,27]. The antioxidant activity of Jamaican cherry fruit has been reported in previous study [8], which attributed to the presence of polyphenol, flavonoid, and vitamin C in the fruit [28]. Fermentation with lactic acid bacteria could improve the bioavailability of fruits’ natural constituents [14]. Here we performed four assays for assessing the antioxidant capacity of fermented Jamaican cherry juice (M1–M6), including DPPH radical, ABTS cation radical, FRAP, and ORAC assay (Figure 3).

The highest DPPH scavenging activity was observed in fermented M1 that contains 100% of Jamaican cherry juice, showing 77.81% DPPH activity, comparable to gallic acid (76.77%) and ascorbic acid (85.57%) that were used as positive control in the experiment. The fermented M1 to M3 substrates exhibited enhanced DPPH scavenging activity compared to their respective non-fermented controls; it was 1.7, 1.8, and 2.2 times higher in fermented M1 to M3, respectively (Figure 3A). Overall, fermented M1 to M3 showed more than 50% of radical scavenging activity after 48 h of fermentation using *L. plantarum*. Similar enhancement in DPPH radical scavenging following fermentation has been reported in previous studies [16,23]. The ABTS radical scavenging activity is also elevated after 48 h fermentation. Similar to DPPH activity the highest ABTS scavenging activity was observed in fermented M1 that contains 100% Jamaican cherry juice, showing 69.25% ABTS activity. Gallic acid and ascorbic acid exhibited 75.43% and 73.45% ABTS activity, respectively. Comparing fermented and non-fermented group, significant increase in ABTS activity was observed in the fermented M1 to M4 (*p* ≤ 0.05), where 1.70; 1.46; 1.47; and 1.76-fold increase in ABTS activity was recorded, respectively (Figure 3B). FRAP (Figure 3C) and ORAC (Figure 3D) values also showed a similar trend. M1 and M2 antioxidative activity significantly elevated (*p* ≤ 0.05) after fermentation using *L. plantarum*. The FRAP and ORAC values for both M1 and M2 are 1.3 times higher following 48 h fermentation

Generally, we showed that *L. plantarum* could elevate antioxidant activity of Jamaican cherry juice, additionally we observed that substrate containing 100% (M1) and 80% (M2) Jamaican cherry juice could significantly increase (*p* ≤ 0.05) antioxidant activity in all tested assays. Suggesting that higher juice content in the substrate correlates with higher antioxidative activity. This results was consistent with our previous study about black grape juices fermentation with *L. plantarum* [16]. Several recent studies also showed the enhancement of antioxidant capacity in fruit juices fermented using lactic acid bacteria, such as in kiwifruit [24], dragon fruit [29], apple [30], and bergamot [31]. Fermentation using lactic acid bacteria could increase the concentration of functional components like phenolic and flavonoids compounds through hydrolysis mechanisms [32]. In addition, the pH change during fermentation could also influence the phytochemical structure of antioxidant compounds [33]. Further, changes in pH could alter the activity of various metabolic enzymes and increase the bioavailability of various compounds [16].

### 2.4. Enzyme Inhibition Ability

Controlling postprandial hyperglycemia is an essential part of diabetes treatment. This could be achieved by reducing or inhibiting the absorption of glucose during the digestion step. There are numerous pathways to prevent the absorption of glucose, one strategy is by inhibiting α-glucosidase, α-amylase, and amyloglucosidase activity. These three enzymes are essential hydrolase that could metabolize carbohydrates and regulate the blood sugar level in the human body. Various extracts from bacteria, fungi, and plants, including fruit, leaves, roots, and roots were reported to be able to inhibit diabetes-related enzymes activity [16,34]. Recent study showed that fermentation using probiotic bacteria, including lactic acid bacteria, could increase the antidiabetic potential of blueberry juice [22]. Such study is essential for the development of fermentation based functional beverages. Here we reported α-glucosidase, α-amylase, and amyloglucosidase inhibition by different formulations (M1–M6) of Jamaican cherry juice fermentation using *L. plantarum*. Overall, significant increase in the inhibition of α-glucosidase, α-amylase, and amyloglucosidase activity (*p* < 0.05) was observed in M1 to M4 following fermentation for 48 h (Figure 4). The highest increase in inhibition against α-glucosidase (1.9 times) and amyloglucosidase (1.6 times) was recorded in fermented M2, while the highest increase in inhibition against α-amylase (1.9 times) was observed in fermented M1. Acarbose, a well-known antidiabetic drug, a positive control in this study, showed higher but comparable inhibitory properties against those three enzymes (Figure 4). These results indicate that fermentation of Jamaican cherry fruit juice using *L. plantarum* can improve the inhibitory activities of the juice and their antidiabetic potential. The increase in inhibitory activities of plant extract against diabetes-related enzymes following *L. plantarum* fermentation is in agreement with previous studies [16,22,35]. Further, the cell-free filtrate of *L. plantarum* X1 was also reported to be able to inhibit α-glucosidase activity [36].

Structural changes of indigenous compounds following fermentation might contribute to the inhibition activity. It is reported that fermentation of *Momordica charantia* (bitter melon) juice using *L. plantarum* BET003 results in the transformation of aglycone compounds and improved antidiabetic potential of the fermented juice [37]. In addition, fermentation might enhance the activity various proteolytic enzymes that can inhibit amylase and amyloglucosidase activity, such as protamex, alcalase, neutrase, and flavourzyme [16].

### 2.5. Mass Spectrometry Analysis of M1 Substrate

Our results showed that total phenolic content, antioxidant activities, and inhibitory activity against diabetes-related enzymes was highest in substrate with 100% Jamaican cherry juice. To evaluate whether those activities are correlated with changes in metabolite composition of the fermented juice, we analyzed the metabolite profile of the fermented and unfermented M1 using UHPLC-QTOF-HRMS/MS. The electrospray ionization mode was set into a negative mode, three more peaks were detected in the fermented M1 namely A, B, and C (Figure 5). We then annotate the putative metabolites A, B, and C using molecular formula and the fragmentation pattern against database, aided with spectral library search together with the suggested fragmentation trees using SIRIUS (https://bio.informatik.uni-jena.de/software/sirius/ [accessed 20 June 2020]). We identified the three putative metabolites of fermented Jamaican fruit juice, including gallic acid (A), dihydrokaempferol (B), and 5,7-dihydroxyflavone (C) (Table 1 and Appendix A).

A previous study reported that *L. plantarum* strain CIR1 could produce gallic acid during fermentation. This bacteria could produce tannase, an enzyme that could facilitate bioconversion of tannic acid to gallic acid [38]. *L. plantarum* strain FNCC 0027 used in this study might also produce tannase that can hydrolyze tannic acid presence in Jamaican fruits [8,39]. Recent study reported that fermentation of *Solanum retroflexum* leaf using *L. plantarum* strain 75 could induce the production of various bioactive compounds, such as gallic acid, vanillic, coumaric, ellagic acid, quercetin, and catechin [40]. Fermentation with *L. plantarum* FNCC 0027 also induces the production of bioactive compounds, including gallic acid, dihydrokaempferol, and 5,7-dihydroxyflavone. The production of dihydrokaempferol possibly due to the *L. plantarum* shikimic pathway that can convert flavonoid naringenin presents in Jamaican fruit into dihydrokaempferol [8]. Hydroxylation of naringenin by hydroxylase enzyme known to be produced by *L. plantarum* could produce intermediate compounds from dihydroflavonols class, such as dihydrokaempferol and dihydroxyflavone [41,42,43]. Gallic acid and flavonoids such as dihydrokaempferol and 5,7-dihydroxyflavone is known to have antidiabetic and antioxidant properties [44,45,46]. Thus, their presence in the fermented substrate can influence the functional properties of the Jamaican cherry juice.

## 3. Materials and Methods

### 3.1. Chemical and Reagents

The chemicals and reagents used are p-nitrophenyl-α-D-glucopyranoside (Calbiochem^®^, San Diego, MA, USA), starch (Merck, Darmstadt, Germany), acarbose (Fluka Analytical, Sigma–Aldrich, Laramie, WY, USA), Man Rogosa Sharpe/MRS broth (Sigma–Aldrich, St. Louis, MO, USA), α-glucosidase from *Saccharomyces cerevesiae* (Sigma–Aldrich, St. Louis, MO, USA), α-amylase from *Aspergillus oryzae* (Sigma–Aldrich, St. Louis, MO, USA), amyloglucosidase from *Aspergillus niger* (Sigma–Aldrich, St. Louis, MO, USA), 2,2-Diphenyl-1-picrylhydrazyl (Sigma–Aldrich, St. Louis, MO, USA), 2,4,6-Tris(2-pyridyl)-s-triazine (Fluka Analytical, Sigma–Aldrich, St. Louis, MO, USA), 3,5-dinitrosalicylic acid (Sigma–Aldrich, St. Louis, MO, USA), 2,2′-Azino-bis(3-ethylbenzothiazoline-6-sulfonic acid) (Sigma–Aldrich, St. Louis, MO, USA), Folin–Ciocalteu’s phenol reagent (Merck, Darmstadt, Germany), gallic acid, peroxidase. Other reagents were analytical grade or better.

### 3.2. Bacterial Strain and Fruit Material

*Lactobacillus plantarum* FNCC 0027 was obtained from the culture collection of Biotechnology Laboratory, Graduate School of Biotechnology, Universitas Gadjah Mada, Yogyakarta, Indonesia. Jamaican cherry fruits were collected from the research field at Research Division for Natural Product Technology (BPTBA), the Indonesian Institute of Sciences (LIPI), Yogyakarta, Indonesia in March 2016. The fruits (250 g) were washed and kept at 4 °C before further use. Jamaican cherry juice was produced from the fruit using sterilized-commercial food juicer at room temperature.

### 3.3. Preparation of Inoculant

*L. plantarum* FNCC 0027 were propagated in MRS broth at 37 °C for 48 h under anaerobic conditions. The cells were harvested, pelletized, and re-suspended in sterilized phosphate buffer saline at 7.02 log CFU/mL for inoculation.

### 3.4. Fermentation Procedure

First, Jamaican cherry juice was produced by homogenizing Jamaican cherry fruits with sterile water in a commercial blender at the concentration of 1 g/mL. The homogenized juices were filtered and pasteurized for 3 min at 85 °C [20]. Fermentation was performed in 250 mL sterilized Erlenmeyer. Erlenmeyer was filled with pasteurized Jamaican cherry juice (J) and sterilized MRS broth (M) with a total reaction volume of 100 mL. Six different J to M ratios were used in this study (in a total of 100 mL), including M1 (1:0), M2 (4:1), M3 (3:1), M4 (1:1), M5 (1:4), and M6 (0:1). Each formulation was subjected to two treatments: control non-fermented group and group fermented using L. plantarum (with an initial starter of 7.02 log CFU/mL). All samples were then incubated aerobically at 37 °C, 100 rpm, for 48 h before harvested for subsequent analysis.

### 3.5. Bacterial Cell Separation

To obtain cell-free supernatant, the fermented juices were centrifuged for 15 min at 10,000× *g* [47]. The resulting supernatant was then filtered through a 0.22 µm membrane filter (Millipore, Burlington, MA, USA) and kept at −20 °C. 

### 3.6. Bacterial Viable Count

Viable count was performed at different intervals over the course of 48 h fermentation (0, 6, 12, 18, 24, 30, 36, 42, and 48 h). To perform viable count, 10 sterile 15mL test tubes were filled with 9 mL of sterilized phosphate buffer saline (PBS). Serial 10-fold dilutions in PBS were then prepared in the test tubes, using 1 mL of fermented juice as the starter. Viable count was next performed using pour plate method in duplicate, 1 mL of solution from each dilution series was mixed with 25 mL tempered (47 °C) Plate Count Agar (OXOID^®^, Basingstoke, England). The plate was incubated for 72 h at 30 °C ± 2 °C, and colony numbers was calculated using a colony counter. Plates with 15 to 300 colonies were considered for colony forming unit calculation. The viable colonies were converted into weighted mean colony forming units per milliliter (CFU/mL) using the following Equation (1):N = ∑C/[(n_1_ + 0.1n_2_)d](1)
N is the number of colonies in the plate; ∑C is the sum of plates containing 15 to 300 colonies; n_1_ is the number of plates retained in the first dilution; n_2_ is the number of plates retained in the second dilution; and d is the first dilution factor. The viable colonies were then converted into log CFU/mL.

### 3.7. pH Measurement

The pH of the samples was measured using Eutech PC 700 pH meter (Thermo Scientific, Waltham, MA, USA). The pH measurement was performed at different intervals over the course of 48 h fermentation (0, 6, 12, 18, 24, 30, 36, 42, and 48 h).

### 3.8. Total Phenolic Content Measurement

Total phenolic content was determined using methods developed by Zhou, Wang, Zhang, Yang, Sun, Zhang, and Yang [24], with minor modifications. Briefly, 25 µL of 0.2 N Folin–Ciocalteu phenol reagents was added into 96-well plates containing 5 µL of each supernatant and 195 µL of distilled water. The mixture was then incubated for 6 min at room temperature in darkness, followed by addition of 75 µL of 7% sodium carbonate. The mixture was then incubated further for 2 h at room temperature in darkness. Blank solution was prepared by the same steps as described above except that supernatant was substituted with 5 µL of water. For all samples, the absorbance at 765 nm was recorded using Multiskan^®^ Go microplate spectrophotometer (Thermo Scientific, Vantaa, Finland). Standard curve was produced using Gallic acid, the equation for the standard curve is y = 0.0104x − 0.0159 (R^2^ = 0.9988) where y is the absorbance at 735 nm and x is the concentration of Gallic acid in µg/mL. Total phenol was expressed as µg of Gallic acid equivalent (GAE) per mL of supernatant (µg GAE/mL).

### 3.9. 2,2-Diphenyl-1-picrylhydrazyl (DPPH) Radical Scavenging Activity

The DPPH scavenging assay was performed using 96-well plates following the method by Kosem, et al. [48], with modifications. Briefly, 50 µL of filtered supernatant was mixed with 70 µL of methanol and the absorbance of the pre-plate reading was recorded at 517 nm. About 80 µL of 0.5 mM DPPH solution in methanol was then added to the well. The degree of purple (from DPPH) decolorization to yellow represents the scavenging efficiency of the supernatants. After an incubation period of 30 min at room temperature (25 ± 2 °C) in the darkness, the decrease in the absorbance was recorded at 517 nm using Multiskan^®^ Go microplate spectrophotometer (Thermo Scientific, Vantaa, Finland). Lower absorbance represents higher free radical-scavenging activity. The scavenging activity against DPPH was calculated using following Equation (2):DPPH scavenging rate (%) = [1 − (Abs_1_ − Abs_0_)] × 100%(2)
Abs_0_ was absorbance of control and Abs_1_ was the absorbance in the presence of supernatant.

### 3.10. 2,2′-Azino-bis(3-ethylbenzothiazoline-6-sulfonic acid) (ABTS) Assay

The ABTS assay was performed according to Re et al. [49]. In brief, radical cation was generated by reacting 5 mL of 7 mM ABTS with 5 mL of 2.45 mM of potassium persulfate. The reaction was performed by incubating the mixture for 16 h in the dark. Working solution for radical cation was obtained by diluting the reacted solution at the OD 0.7 ± 0.02 at 734 nm. Subsequently, 300 μL of the diluted radical cation solution was mixed with 3 μL of supernatant. The ODs was then recorded at 734 nm after 10 min incubation at room temperature using Multiskan^®^ Go microplate spectrophotometer (Thermo Scientific, Vantaa, Finland). The ABTS radical scavenging activity was calculated using Equation (2).

### 3.11. Ferric Reducing-Antioxidant Power (FRAP) Assay

The FRAP assay was conducted according to Cecchini and Fazio [50], which was originally described by Benzie and Strain [51], with minor modifications. In brief, FRAP reagent was prepared by mixing 300 mM sodium acetate buffer (pH 3.6), 10 mM of 2,4,6-Tris(2-pyridyl)-s-triazine (TPTZ) in 40 mM HCl, and 20 mM FeCl_3_.6H_2_O at 10:1:1 (*v/v/v*) ratio. Subsequently, 10 μL of supernatant was added to the 96-well plate containing 300 μL FRAP solution as an oxidizing reagent. After 5 min incubation in dark at 37 °C, the absorbance was measured at 593 nm using Tecan Infinite^®^ 200 Pro microplate reader. The experiment was calibrated with FeSO_4_·7H_2_O and the results were expressed in terms of FeSO_4_·7H_2_O equivalents (μM).

### 3.12. Oxygen Radical Absorbance Capacity (ORAC) Assay

ORAC were measured according to Zulueta, et al. [52]. The 2,2′-Azobis(2-amidinopropane) dihydrochloride (AAPH) radical stock solution was prepared freshly by adding 434 mg of AAPH to 10 mL of phosphate buffer (75 mM) to obtain final concentration of 161 mmol/L. Fluorescein stock solution (1.03 mmol/L) was prepared using phosphate buffer solution. Subsequently, the supernatant (25 μL) was added to the 96-well plate and mixed with 150 μL of fluorescein solution (40 nm/L). Mixture was then incubated for 5 min at 37 °C. Subsequently, 25 μL AAPH solutions was added and the fluorescein was recorded immediately at an excitation wavelength of 485 nm and emission wavelength of 535 nm. The fluorescein was monitored using Tecan Infinite^®^ 200 Pro microplate reader every minute for 30 min, ORAC values were expressed in term of Trolox equivalents (μM).

### 3.13. In Vitro Inhibiting Activity of α-Glucosidase

The α-glucosidase inhibitory activity of the fermented supernatant was determined according to the chromogenic method, in a 96-well plate, according to our previous published method [16]. First, 20 µL of supernatant was mixed with 10 µL of a 1.0 U/mL α-glucosidase and 50 µL of 0.1 M sodium phosphate buffer (pH 6.9). The mixed solution was incubated at 37 °C for 15 min. After pre-incubation, the enzymatic reaction was initiated by adding 20 µL of 5 mM p-nitrophenyl-α-D-glucopyranoside solution in 0.1 M sodium phosphate buffer (pH 6.9). The mixture was incubated for 20 min at 37 °C. The absorbance was subsequently measured at 405 nm using Multiskan^®^ Go microplate spectrophotometer (Thermo Scientific, Vantaa, Finland). Percent inhibition was calculated relative to the diabetes drug acarbose as the reference. Reaction system without supernatant was used as negative control while reaction system without α-glucosidase was used as a blank for correcting the background absorbance. The percentage of enzymatic inhibition activity was calculated using following Equation (3):% inhibition activity = [(Abs_A_ − Abs_B_)/Abs_A_] × 100%(3)
Abs_A_ is the absorbance of the control and Abs_B_ is the absorbance of the tested supernatant.

### 3.14. In Vitro Inhibiting Activity of α-Amylase

The α-amylase inhibitory activities of the fermented supernatant were carried out according to procedure reported by Telagari and Hullatti [53]. The assay system was carried out in a 96-well plate. First, a reaction mixture containing 10 μL of a 2.0 unit/mL α–amylase, 50 μL sodium phosphate buffer (0.1 M, pH = 6.9), and 20 µL of supernatant was prepared. The mixed solution was incubated at 37 °C for 20 min. After pre-incubation, 50 μL of 1% soluble starch (Merck, Darmstadt, Germany) in 0.1 M sodium phosphate buffer pH 6.9 was added as a substrate and the mixture was incubated further for 30 min at 37 °C. Next, 100 μL of the 3.5-dinitrosalicylic acid solution was added and heated at 100 °C in a water bath for 10 min. The absorbance was subsequently measured at 540 nm using Multiskan^®^ Go microplate spectrophotometer (Thermo Scientific, Vantaa, Finland). Acarbose was used as a positive reference standard. The percentage of α-amylase inhibition was calculated using Equation (2).

### 3.15. In Vitro Inhibiting Activity of Amyloglucosidase

The amyloglucosidase (exo-1,4-α-glucosidase) inhibitory activities of the fermented supernatant was carried out in 96-well plate according to Saul et al. [54] and Warren et al. [55], with modifications. First, a mixture containing 10 µL of a 1.0 U/mL α-amyloglucosidase, 10 µL of 0.1 M sodium acetate (pH 5.0), and 25 µL of supernatant was prepared. The mixed solution was incubated at 37 °C for 20 min. After pre-incubation, the enzymatic reaction was initiated by adding 5 µL of 5 mM p-nitrophenyl-α-D-glucopyranoside solution in 0.1 M sodium phosphate buffer (pH 6.9). At the end of the incubation, 200 µL of 0.4 mM glycine buffer (pH 10.4) was added to each well to stop reaction. The p-nitrophenil released was then measured at 410 nm using Tecan Infinite^®^ 200 Pro microplate reader.

### 3.16. UPLC-HRMS-MS Analysis

High resolution MS was carried out using Bruker maXis 4G ESI time of flight mass-spectrometer (Bruker Daltonics, Bremen, Germany) attached to an Ultimate 3000 HPLC (Thermo Fisher Scientific, Bremen, Germany). The UHPLC-method was performed using Reprosil 3 µm C18 100 Å, 10 × 3.3 mm (flow rate of 0.3 mL/min, monitored at 210 and 240 nm) with linear gradient starts from 90% to 0% of A (A: ddH_2_O, B: Acetonitrile, both solvents containing 0.01% formic acid) for 30 min and held constant of 100% B for 10 min. The parameter was set in a capillary voltage of 4500 V, nebulizer nitrogen pressure of 2 bars, the dry gas flow of 9 L/min source temperature, ion source temperature 200 °C, and spectral rate of 3 Hz. The MS data subsequently analyzed using Bruker Compass Data Analysis 4.4 SR1(x64).

### 3.17. Statistical Analysis

All assays were conducted in triplicate. The mean and standard error (mean ± SE) was determined of each data. The analysis of variance (ANOVA) and student t-test was employed using SPSS 16 to determine the level of statistical differences between control and Jamaican juice formulation fermented with *L. plantarum*. Differences at *p* < 0.05 were considered statistically significant.

## 4. Conclusions

Our results demonstrated that fermentation of Jamaican cherry juice using *L. plantarum* FNCC 0027 could significantly improve the functional activities of the substrates, including higher polyphenol content, antioxidant capacity (DPPH, ABTS, FRAP, and ORAC), and inhibitory activity against diabetic-related enzymes (α-glucosidase, α-amylase, and amyloglucosidase). Various fermentation formulations with different ratios of Jamaican cherry juice and MRSB volume were studied in this study and we found that M1 formulation with 100% Jamaican cherry juice showed the highest functional activities compared to the other composition. This result indicates that the addition of MRSB is not required to facilitate effective fermentation and improvement of Jamaican cherry juice functional properties. In summary, the functional properties and beneficial activity of Jamaican cherry juice can be enhanced by fermentation with *L. plantarum*, one of “generally regarded as safe” (GRAS) strain. This work can be used as the basis for the development of new functional beverages using fermented Jamaican cherry juice as the main component.

## Figures and Tables

**Figure 1 molecules-26-02868-f001:**
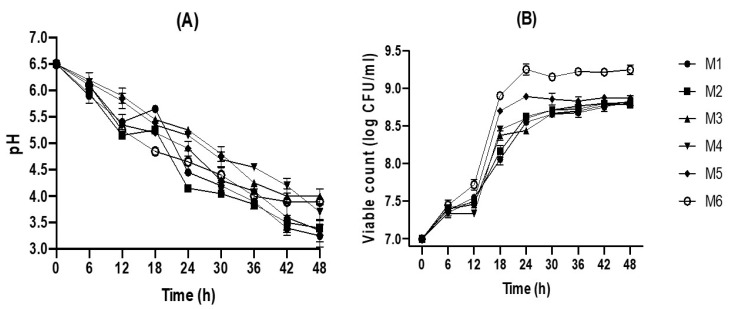
Effect of fermentation on pH and viable count (**A**) the change of pH (**B**) viable count.

**Figure 2 molecules-26-02868-f002:**
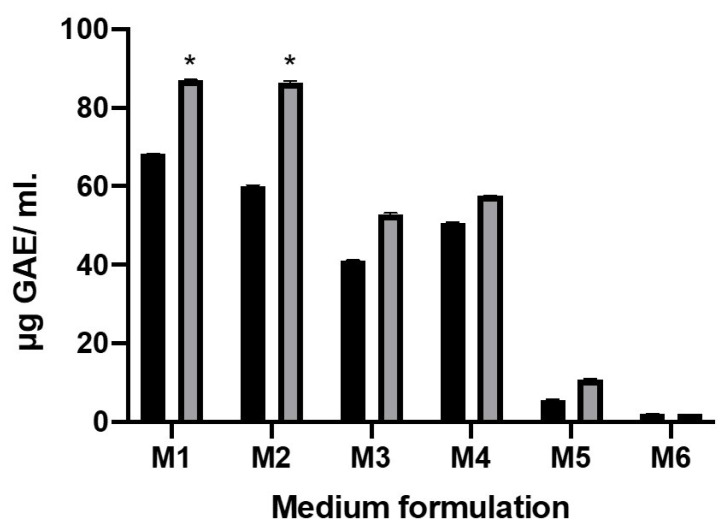
Effect fermentation of *L. plantarum* on the total phenol content. Black bar represent phenol content before fermentation and gray bar represent phenol content after fermentation. Each value represents the mean ± SE (n = 3). The star (*) means significantly different at *p* < 0.05.

**Figure 3 molecules-26-02868-f003:**
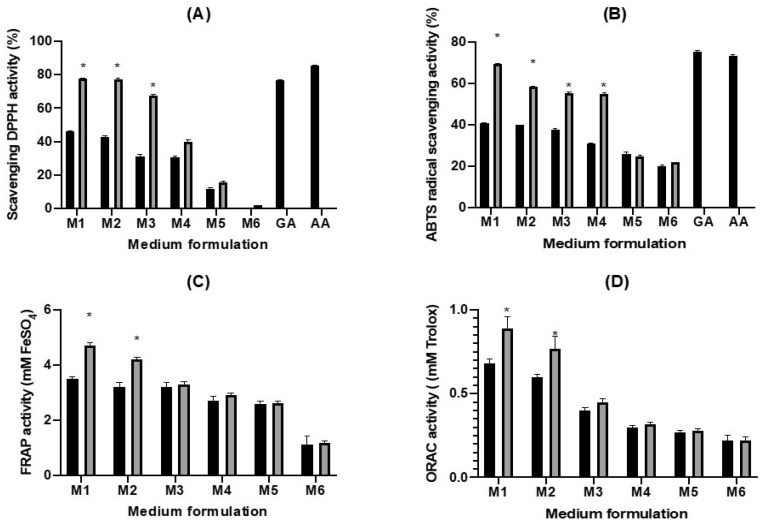
The effect of fermentation by *L. plantarum* on the antioxidant activities of Jamaican cheery juices. (**A**) Scavenging DPPH activity (**B**) ABTS Scavenging radical activity (**C**) FRAP activity (**D**) ORAC activity. Black bar represent antioxidant activities before fermentation and gray bar represent antioxidant activities after fermentation. Each value represents the mean ± SE (n = 3). The star (*) means significantly different at *p* < 0.05.

**Figure 4 molecules-26-02868-f004:**
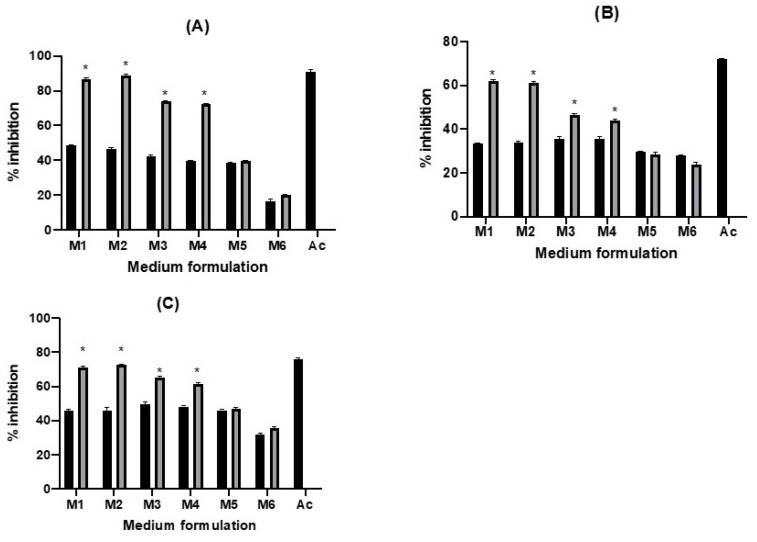
Effect fermentation of *L. plantarum* to the inhibition of (**A**) α-glucosidase, (**B**) α-amylase, and (**C**) amyloglucosidase. Black bar represent inhibitory activities before fermentation and gray bar represent inhibitory activities after fermentation M1 to M6 means medium formulation and Ac means Acarbose 5 mg/mL. Each value represents the mean ± SE (n = 3). The star (*) means significantly different at *p* < 0.05.

**Figure 5 molecules-26-02868-f005:**
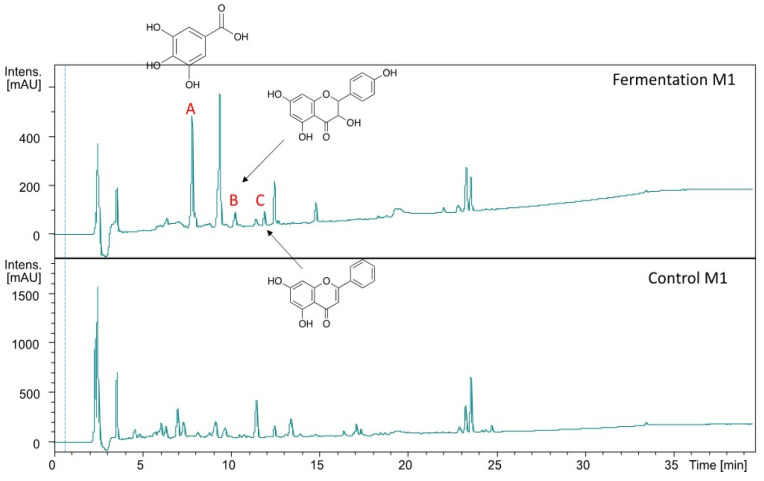
The UV chromatogram of M1 containing *L. plantarum* (**above**) and M1 without *L. plantarum* (**below**).

**Table 1 molecules-26-02868-t001:** Chemical constituent identified after fermentation of M1 using *L. plantarum*.

Peak	Putative Compound	Experiment (*m*/*z*)	Theoretical (*m*/*z*)	Adduct	Error (ppm)	rdb
A	gallic acid	169.0141	169.0142	[M-H]^−^	0.7	5
B	dihydrokaempferol	287.0561	287.0561	[M-H]^−^	−0.1	10
C	5,7-dihydroxyflavone	253.0505	23.0506	[M-H]^−^	0.5	11

rdb means ring double bond.

## Data Availability

Publicly available datasets were analyzed in this study. This data can be found at: https://www.kaggle.com/microbiologii/dataset-plantarum (accessed on 20 June 2020).

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
