# Peer review of "Fermentation of Jamaican Cherries Juice Using Lactobacillus plantarum Elevates Antioxidant Potential and Inhibitory Activity against Type II Diabetes-Related Enzymes"

_molecules, 2021, doi:10.3390/molecules26102868_

Round 1

Reviewer 1 Report

The manuscript from Frediansyah et al. describes the use of lactic fermentation to improve the functional characteristics of Jamaican cherries juice (Muntingia calabura Linn.). The authros applied a strain of Lactobacillus plantarum for this end.

The paper is well organized and interesting. The results are clearly presented, as expected for a manuscript published in Molecules.

Some minor issues are pointed in the file attached and should be solved before the final acceptance of this manuscript. 

Author Response

Thank you for your encouraging words and suggestions. We are pleased to know that the first reviewer thought the paper was well-organized and interesting, making it appropriate for publication in Molecules. Thank you for your kind words.

We also appreciate your suggestions for text revision. We apologize for incorrectly formatting the species names and equations. The suggested changes have already been incorporated into the revised manuscript. We have attached the revised manuscript. Revisions made in response to first reviewer are highlighted in red, while revisions made in response to the second reviewer are highlighted in blue.

Reviewer 2 Report

The authors present interesting results that fit into the application of fermentation processes to increase the bioactivity of plant materials. 

The work requires some necessary corrections :

  1. Please use italics for all latin names in text.
  2. Line 23 please add strain number
  3. Line 20 - it should be "probiotic bacteria"
  4. Double spaces in some lines - 20, 21, ...
  5. Line 40 - it should be "phenolic compounds"
  6. I did not find microbiological analyses and acidity test methodologies in the methods section.
  7. The authors mention that the pH affects the taste of the final product - were sensory analyzes carried out? Another issue is the use of MRS broth in fermentation systems with juice - can it have a positive / negative effect on the taste of the product? Similarly, when it comes to antioxidant potential - it's rather obvious that a juice rich in polyphenols and flavonoids will have more activity than MRS broth - so is it right to mix natural juice with a microbial medium?  Please comment. 
  8. Line 101 - please add units.
  9.  line 102 - consonance?
  10. Lines 227/228 - some problems with references
  11. in whole text please use mL instead of ml
  12. section 3.4 is unclear - what does it mean 1 g/ml of cherry juice? Were the juice / MRS systems incubated aerobically or anaerobically? 
  13. 3.7 - 2,2-Diphenyl....
  14. 3.8 - 2,2'-azino-bis...
  15. Lines 315, 319, 320 - please correct chemical formula and use subscript
  16.  α or alpha - please unify

Author Response

We appreciate the comments made by the two reviewers on our manuscript. Their comments are helpful in terms of improving our manuscript and serving as a beneficial guide for our research. We have revised the manuscript in accordance to their comments and suggestions. The revised section is highlighted in blue and red colour in the manuscript. The following sections contain the manuscript's major corrections and responses to the reviewer’s comments:

  1. Please use italics for all Latin names in the text.

Response: We apologize for incorrectly formatting some of the species’ names, we have revised the manuscript in accordance to the reviewer's comments.

  1. Line 23 please add strain number

Response: Thank you for the correction, we have updated the manuscript by adding FNCC 0027.

  1. Line 20 - it should be "probiotic bacteria"

Response: Thank you for the correction, we have revised the manuscript in accordance to the reviewer's comments.

  1. Double spaces in some lines - 20, 21, ...

Response: Thank you for the correction, we have revised the manuscript in accordance to the reviewer's comments.

  1. Line 40 - it should be "phenolic compounds"

Response: Thank you for the correction, we have revised the manuscript in accordance to the reviewer's comments.

  1. I did not find microbiological analyses and acidity test methodologies in the methods section.

Response: We apologize that we have not include the method for bacterial viable counts and pH measurement. We appreciate the reviewer's suggestion to improve the method section. We have updated the sections with detailed method for bacterial viable counts and pH measurement that we performed on the research. The revised section of the manuscript is highlighted in blue for the section 3.6. Bacterial viable count and 3.7. pH measurement (Line 277-297)

  1. a. The authors mention that the pH affects the taste of the final product - were sensory analyzes carried out?

Response: We did not conduct sensory analysis in a formal manner. We are citing paper published by McFeeters, (Fermentation microorganisms and flavor changes in fermented foods. Food science and technology 5042004,69, (1), FMS35-FMS37), they mentioned pH's effect on flavor. We modified the sentences to make it clear that it is based on previous report and not from our own experiment (Line 88 to 90)..

        7.b. Another issue is the use of MRS broth in fermentation systems with                 juice - can it have a positive / negative effect on the taste of the product?

Response: Because the MRS broth contains peptone, yeast extract, glucose, and Sorbitan mono-oleate, it might have noticeable effect on the taste of the product at high concentrations such as in M4 to M6 substrate. MRS broth was effective in promoting the growth of L. plantarum FNCC 0027. We want to evaluate the effect of different L. plantarum FNCC 0027 concentration in the substrate (due to different MRS broth ratio, Figure 1B) of the fermentation and functional properties of the juices. From our study, formulation with 100% Jamaican cherry juice (M1) exhibited highest antioxidant and anti-diabetic activity in vitro compared to other compositions (M2-M6), suggesting that higher L. plantarum FNCC 0027 concentration does not improve the functional properties of the juices. Thus, it is advised to ferment the entire juice only with L. plantarum without MRS broth addition. We revising the conclusion section to emphasize this finding.

         7.c. Similarly, when it comes to antioxidant potential - it's rather obvious               that a juice rich in polyphenols and flavonoids will have more activity                   than MRS broth - so is it right to mix natural juice with a microbial                       medium?  Please comment.

Response: Previously it is unknown whether L. plantarum FNCC 0027 could survive and facilitate fermentation in 100% Jamaican cherry juice. MRS broth is a medium that promotes the growth of lactobacilli in general, and specifically L. plantarum. Therefore, we formulate different composition of Jamaican cherry juice mixed with MRS broth and measure the viable count of L. plantarum in these various composition at different time interval. We want to evaluate whether the addition of MRS broth could enhance L. plantarum growth and facilitate a better fermentation process, that in turn enable higher accumulation of polyphenols and flavonoids in the fermented juice. If we found that the addition of MRS broth could enhance the fermented juice functional properties we plan to develop low-cost food grade MRS broth. Nevertheless, we found that L. plantarum FNCC 0027 could effectively ferment the juice without the addition of MRS broth. Therefore, direct fermentation of 100% juice without MRS broth is the most suitable substrate for further development into a functional beverage.

  1. Line 101 - please add units

Response: We revised the manuscript with a blue mark in response to the reviewer's comments (by adding folds)

  1. line 102 - sonsonance?

Response: Thank you for the correction, we revised the manuscript with a blue mark in response to the reviewer's comments (yes it should be consonance)

  1. Lines 227/228 - some problems with references

Response: We appreciate the reviewer's patience in assisting us in improving our article. In response to the reviewer's suggestions, we thoroughly checked and rewrote the references (References 41 to 43).

  1. in whole text please use mL instead of ml

Response: We revised the manuscript with a blue mark in response to the reviewer's comments, we use mL throughout the manuscript.

  1. section 3.4 is unclear - what does it mean 1 g/ml of cherry juice? Were the juice / MRS systems incubated aerobically or anaerobically?

Response: In response to the reviewer's suggestions, we thoroughly checked and rewrote section 3.4. Fermentation procedure. To ensure that reader could understand the sample preparation and fermentation procedure. The manuscript's revised section is highlighted in blue (Line 261 to 271).

  1. 7 - 2,2-Diphenyl....

Response: We revised the manuscript with a blue mark in response to the reviewer's comments (Line 244)

  1. 8 - 2,2'-azino-bis…

Response: We revised the manuscript with a blue mark in response to the reviewer's comments (Line 246)

  1. Lines 315, 319, 320 - please correct chemical formula and use subscript

Response: We revised the manuscript with a blue mark in response to the reviewer's comments (Line 336 to 343)

  1. α or alpha - please unify

Response: We revised the manuscript with a blue mark in response to the reviewer's comments and we unified with α

Round 2

Reviewer 2 Report

I reanalyzed the document and the answers given by the authors, and consider that the document was clearly improved, thus it can be accepted. Congratulations.